# Artificial Intelligence in Breast Ultrasound: From Diagnosis to Prognosis—A Rapid Review

**DOI:** 10.3390/diagnostics13010058

**Published:** 2022-12-26

**Authors:** Nicole Brunetti, Massimo Calabrese, Carlo Martinoli, Alberto Stefano Tagliafico

**Affiliations:** 1Department of Experimental Medicine (DIMES), University of Genova, Via L.B. Alberti 2, 16132 Genoa, Italy; 2Department of Radiology, IRCCS—Ospedale Policlinico San Martino, Largo Rosanna Benzi 10, 16132 Genoa, Italy; 3Department of Health Sciences (DISSAL), University of Genova, Via L.B. Alberti 2, 16132 Genoa, Italy

**Keywords:** artificial intelligence, breast cancer, ultrasound, deep learning, machine learning

## Abstract

Background: Ultrasound (US) is a fundamental diagnostic tool in breast imaging. However, US remains an operator-dependent examination. Research into and the application of artificial intelligence (AI) in breast US are increasing. The aim of this rapid review was to assess the current development of US-based artificial intelligence in the field of breast cancer. Methods: Two investigators with experience in medical research performed literature searching and data extraction on PubMed. The studies included in this rapid review evaluated the role of artificial intelligence concerning BC diagnosis, prognosis, molecular subtypes of breast cancer, axillary lymph node status, and the response to neoadjuvant chemotherapy. The mean values of sensitivity, specificity, and AUC were calculated for the main study categories with a meta-analytical approach. Results: A total of 58 main studies, all published after 2017, were included. Only 9/58 studies were prospective (15.5%); 13/58 studies (22.4%) used an ML approach. The vast majority (77.6%) used DL systems. Most studies were conducted for the diagnosis or classification of BC (55.1%). At present, all the included studies showed that AI has excellent performance in breast cancer diagnosis, prognosis, and treatment strategy. Conclusions: US-based AI has great potential and research value in the field of breast cancer diagnosis, treatment, and prognosis. More prospective and multicenter studies are needed to assess the potential impact of AI in breast ultrasound.

## 1. Introduction

Breast cancer (BC) is the most common malignancy in women and the second leading cause of cancer death, so the early diagnosis of BC remains crucial [1]. Mammography is the first-line examination for breast cancer screening. In several settings, another fundamental tool is represented by breast ultrasound (US). In particular, US is recommended as a first-line examination in young women, during pregnancy or breastfeeding, and as an additional examination in women with dense breasts after mammography [2]. US presents several advantages, including safety, low cost, rapid execution, and cost-effectiveness. However, US remains an operator-dependent examination. For these reasons, in recent years, there has been a strong development of artificial intelligence (AI) in breast ultrasound. AI systems involve three working steps: image processing, segmentation, and feature extraction. AI is used to provide predictive models based on the analysis of the features extracted from radiological data. The first step is represented by lesion detection and segmentation (unsupervised or supervised). Then, radiomic analysis is performed with biomarker extraction and analysis used to obtain information for diagnosis or prognosis, as will be explained later. AI systems process a large amount of iconographic data from different imaging modalities to obtain output information [3]. In this framework, we can divide these into two main broad approaches:

Machine learning (ML)—a complex multistep process that uses texture analysis to extract quantitative information from radiological images to create prediction models and decision support tools.

Deep learning (DL)—represents an evolution of ML. The system is able to extrapolate inputs directly from images. These inputs come directly to a multilayer neural network. It does not require manual design features required by traditional methods but automatically learns features [4].

Currently, radiomics is a complex process that involves several steps, and generally, its application is not fully automated. Unsupervised machine learning does not need the training phase to work, but it is typically used just for classification purposes. Supervised machine learning has more general applications, for example, for regression and prediction. Deep learning is an extreme modification of machine learning. AI looks promising in diagnosing breast lesions, predicting molecular subtypes of breast cancer, evaluating axillary lymph node status, and evaluating the response to neoadjuvant chemotherapy, as demonstrated by several studies in the literature. The aim of this rapid review was to assess the current development and research status of US-based AI in the field of breast cancer.

## 2. Materials and Methods

### 2.1. Identification of Studies

Two investigators with experience in medical research performed literature searching and data extraction on PubMed. The two researchers have had extensive experience in medical research and systematic reviews (N.B. and A.T.), with specific expertise in breast cancer imaging, ultrasound imaging, and radiomics. Literature search on PubMed was conducted with the following search strategy:

(((“diagnostic imaging”(MeSH Subheading) OR (“diagnostic”(All Fields) AND “imaging”(All Fields)) OR “diagnostic imaging”(All Fields) OR “ultrasound”(All Fields) OR “ultrasonography”(MeSH Terms) OR “ultrasonography”(All Fields) OR “ultrasonics”(MeSH Terms) OR “ultrasonics”(All Fields) OR “ultrasounds”(All Fields) OR “ultrasound s”(All Fields)) AND (“breast”(MeSH Terms) OR “breast”(All Fields) OR “breasts”(All Fields) OR “breast s”(All Fields)) AND (“artificial intelligence”(MeSH Terms) OR (“artificial”(All Fields) AND “intelligence”(All Fields)) OR “artificial intelligence”(All Fields)))) AND (2017:2023(pdat)).

Only publications after 2017 and in the English language were considered. Reports or case series, review articles, letters, comments, or studies with incomplete data were excluded. The studies included in this rapid review evaluated the role of artificial intelligence in breast ultrasound providing data on BC diagnosis, prognosis, or BC staging. In Figure 1, a schematic representation of a typical workflow of data extraction from US images is presented. After the segmentation is completed, a radiomic analysis is carried out, extracting quantitative features from the obtained volumes. The extracted data are processed by software for possible clinical use, such as diagnosis, reduction in biopsies number, or as prognostic tools. We only used data available in the published studies without contacting authors.

### 2.2. Studies Examination and Data Extraction

Two authors independently extracted the data from the eligible study. Fifty-eight main studies were identified, as reported in Figure 2. Discrepancies were resolved by discussion between two authors. From each study, we extracted the following data: first author, publication year, design of the study (retrospective or prospective, single center or multicenter), study population, aim of the study, imaging modality, AI modality, test data set, and training-validation dataset. Papers without solid statistical analysis were excluded from the study. Then, we performed a descriptive analysis of these studies.

## 3. Results

A total of 58 main studies, all published after 2017, were included. Only 9/58 studies were prospective (15.5%), and 13/58 studies (22.4%) used an ML approach. The vast majority (77.6%) used DL systems. Studies performed with the aim of evaluating the role of AI in the diagnosis of BC was 32/58 (55.1%), assessing lymph node status was 8/58 (13.8%), predicting the response to neoadjuvant chemotherapy (NAC) was 6/58 (10.3%), and predicting molecular subtypes of BC was 8/58 (13.8%). Only four studies investigated the role of AI in the upstage of ductal carcinoma in situ (DCIS) to invasive ductal carcinoma (IDC) and in the prediction of the BC prognosis. Only 8/58 studies (13.7%) also used shear wave imaging (SWE)/quantitative ultrasound (QUS) and color doppler flow imaging (CDFI) in addition to B-mode US.

Table 1, Table 2, Table 3 and Table 4 show all the included studies, their aims, and the performance of the AI algorithms.

The mean values of sensitivity (true positive rate), specificity (true negative rate) and AUC (area under the curve) were calculated for the main study categories with a meta-analytical approach.

For the BC diagnosis:the average sensitivity value was 84.00 (95% CI 73.85–94.15);the average specificity value was 85.67 (95% CI 73.41–97.92);the average AUC value was 90.6429 (95% CI 88.4898–92.7959).

For the assessment of lymph node status:the average sensitivity value was 66.475 (95% CI 31.3442–101.6058);the average specificity value was 66.455 (95% CI 31.6272–101.2828);the average AUC value was 84.5714 (95% CI 76.8547–92.2882).

For prediction of the response to neoadjuvant chemotherapy:the average sensitivity value was 85.3333 (95% CI 76.0254–94.6412);the average specificity value was 78.1667 (95% CI 61.6080–94.7254);the average AUC value was 85.8333 (95% CI 77.5059–94.1608).

For prediction of the molecular subtypes of BC:the average sensitivity value was 88.00 (95% CI 79.1947–96.8053);the average specificity value was 82.00 (95% CI 69.8337–94.8329);the average AUC value was 86.875 (95% CI 81.4489–92.3011).

## 4. Discussion

### 4.1. Diagnosis of Breast Cancer

US represents a fundamental tool to evaluate and characterize breast masses after mammography. Interpreting breast ultrasound is challenging. According to BI-RADS [59], many features have to be analyzed to evaluate a breast nodule, including size, shape, margin, echogenicity, posterior acoustic features, and orientation. Diagnosis depends on the experience of radiologists, and thus, great interobserver variability may happen. The malignancy risk of BI-RADS-4 lesions covers a range from 2% to 95%. After breast US, 5–15% of patients are recalled, and 4–8% of these undergo biopsy with the problem of false positive examinations [2]. One of the advantages of the AI system is the possibility of detecting ultrasound features that cannot be distinguished by the human eye. Especially in BI-RADS 4 breast lesions, AI could help to reduce the rate of unnecessary biopsies. When AI is present in the clinical workflow of radiologists, it can be used as a “second opinion” in the assessment of suspicious breast nodules [60]. Several studies have assessed the value of an AI system in the decision-making process (downgrading and upgrading). According to Lyu et al. [8], an adjusted BI-RADS classification after using AI could lead to a decreased biopsy rate (from 100% to 67.29% in this study), with a great reduction in unnecessary biopsies. As also demonstrated by Wang et al. [11], after hypothetically downgrading with AI, 14 samples would have been avoided without cancer missed. Shen et al.’s paper [12], based on a large amount of data, showed how, helped by artificial intelligence, radiologists could decrease false positive rates by 37.3% and reduce requested biopsies by 27.8%, maintaining the same sensitivity. Gu et al. [27], in an important multicenter study with a large training cohort, internal test cohort, and external test cohorts, demonstrated that a DL system presented a similar performance with respect to the expert and had a significantly higher performance than that of inexperienced radiologists. AI could also have a fundamental role in reducing the US interpretation time, as demonstrated by Lai et al.’s study [34], in which it was reduced by approximately 40% due to AI system support.

Concluding, at present, many studies have shown that AI has excellent diagnostic performance in breast cancer diagnosis [61,62]. However, most studies are only single-center studies, and in most cases, there was no independent external test set.

### 4.2. Prediction of Molecular Subtypes of Breast Cancer

As known, there are four different molecular subtypes of breast cancer: luminal A, luminal B, HER-2 overexpression, and triple-negative breast cancer (TNBC). Each subtype presents different biological characteristics, imaging characteristics, prognosis, and treatment [63]. Molecular subtypes, in fact, can affect the response to NAC; HER2+ and TNBC cancers have a higher probability of responding well to preoperative therapy, but, on the other hand, patients with nonluminal disease have worse recurrence-free and disease-specific survival [64]. Therefore, it is essential to make a histological and molecular diagnosis before surgery. Currently, the gold standard is the microhistological (core or vacuum) biopsy. One of the main limitations of pathological biopsy is the possible underestimation of the sample because of the heterogeneity of the tumor, and for this reason, the biopsy may sometimes not be representative of the cancer [65]. In the literature, there are several studies based on the prediction of breast cancer molecular subtypes based on the AI US system. Ma et al. [42] developed an ML system to differentiate luminal/HER 2+/TNBC/versus other subtypes, ER-positive versus ER-negative, HER2-positive versus HER2-negative, and high Ki-67 versus low Ki-67 expression, proving that a machine learning model can assist radiologists to evaluate the molecular subtype of breast cancer. Other studies, on the other hand, have focused only on the diagnosis of TN subtypes. Several retrospective studies showed, in fact, that TNBC was more likely to show benign features, such as an oval or round shape or smooth or circumscribed margins, and was less likely to have an echogenic halo [66]. All these characteristics may lead to a diagnostic delay, resulting in a poor outcome. Therefore, the AI-based diagnosis of TN breast cancer is crucial in clinical diagnosis and treatment. Every study included in this paper showed good diagnostic performance in the diagnosis of TNBC versus other histological subtypes. Zhou et al. [41] established a multimodal approach based on US, SWE, and CDFI to predict the subtype of breast cancer. This multimodal AI system performed better than US alone (AUC: 0.89–0.96 vs. 0.81–0.84) and, surprisingly, also performed better than the core needle biopsy (AUC: 0.89–0.99 vs. 0.67–0.82, *p* < 0.05). In clinical practice, a possible mismatch between the biopsy result and the AI algorithm could lead to a rebiopsy with the sample at another site. This model also achieved excellent results (AUC: 0.934–0.970) in differentiating TNBC and non-TNBC. AI has the potential to provide a noninvasive method of assessing tumor biology before surgical or medical treatment.

### 4.3. Prediction of Axillary Lymph Node Metastases in Breast Cancer

Identification of lymph node metastasis in breast cancer is crucial for correct diagnostic and therapeutic planning. US has a fundamental role in determining axillary lymph node (ALN) status. The prediction of preoperative lymph node metastases can provide valuable information for determining adjuvant therapy and making surgical plans (lymph node dissection) that are often associated with complications, such as lymphedema. Several studies have described a great deal of breast US characteristics associated with lymph node metastasis and, in addition, lymphatic invasion and the size of the breast cancer are associated with the presence of metastatic cells [67]. At US examination, lymph node metastasis is characterized by unclear margins, irregular shapes, and loss of fatty hilum, but lymph nodes with micrometastases are missed [68]. In addition, all these evaluations are based on expertise and experience, which are operator-dependent. In all the analyzed studies, AI algorithms improved metastatic lymph node detection compared with standard radiological evaluation. The studies mainly focused on predicting the presence or absence of ALN metastasis, but a maximum value of sensitivity below 0.7 could be considered promising but relatively low to enter into clinical practice soon. We acknowledge that differences in cut-off values and reference standards across the different studies involved could influence the results. In addition, we do not know to what extent the relatively low specificity of US in predicting LN metastasis could lead to overtreatment or undertreatment. Prospective, well-designed, and possibly multicentric studies are needed to clarify the role of US in LN metastasis prediction. Guo et al. [50] analyzed 3049 US images of 937 patients and developed a DL radiomics-based prediction model to assess the risk of metastasis of sentinel lymph nodes (SLNs) (AUC = 0.84, sensitivity = 98.4%) and nonsentinel lymph nodes (NSLNs) (AUC = 0.81, sensitivity = 98.4%). Zhou et al. [47] used three different AI algorithms, and all three performed better than the medical radiologist (the sensitivity and specificity of radiologists were 73% and 63%). Other studies also predicted the metastatic burden of ALNs. Zheng et al. [45] showed good results in predicting 1–2 (low metastatic burden) or ≥3 (heavy metastatic burden) ALN metastasis (AUC = 0.90). The prediction of lymph node metastasis by combining the US characteristics of the tumor, US characteristics of the lymph node, and artificial intelligence features could lead to a great diagnostic effect in clinical practice if diagnostic yield above standard radiological evaluation is at least equal to standard sentinel lymph node assessment.

### 4.4. Prediction Response to NAC in Breast Cancer

According to several guidelines [69,70], neoadjuvant chemotherapy (NAC) is the standard of care for patients with locally advanced breast cancer (LABC), biologically aggressive tumors, or nonsurgical patients. LABC is defined as a tumor greater than 5 cm and with skin and/or chest wall involvement or a tumor with many metastatic lymph nodes [71]. Pathological complete response (pCR) is associated with better clinical outcomes compared with nonresponder or partial responder patients. Combining US imaging technology with AI to extract quantitative information from US images can provide more objective information about the response to neoadjuvant chemotherapy. All the analyzed studies give good results in predicting the response to therapy. Jiang et al. [54] used AI to establish a pCR prediction model based on breast cancer US images before and after neoadjuvant chemotherapy in locally advanced breast cancer. The model had a good predictive value (AUC = 0.94). Gu et al. [55] used US images at different NAC time points to establish a NAC response (before, after the second course, and after the fourth course of NAC), with an AUC value that increased during the course of therapy (AUC of 0.812 after the second courses and an AUC of 0.937 after the fourth courses). Gu et al. [55] also identified 90.5% of nonresponder patients. Jiang et al. [54] also demonstrated that prediction within the hormone receptor–positive/human epidermal growth factor receptor 2 (HER2)–negative, HER2+, and triple-negative subgroups also achieved good discrimination performance, with an AUC of 0.90, 0.95, and 0.93. The prediction of the NAC response before starting treatment could be extremely useful for clinicians for risk stratification and targeting treatment, and in future, this could lead to precision medicine permitting therapy adjustments.

### 4.5. Upstage of DCIS in IDC

A correct histological diagnosis of BC before surgery is fundamental. In fact, sentinel lymph node biopsy is not usually performed in the case of partial mastectomy for DCIS [72]. This patient could undergo SLN biopsy if DCIS is upgraded postoperatively to invasive cancer. This could lead to a second surgery, with increased costs for the health system. One of the main limitations of microhistological biopsy is the possible underestimation of the sample because of the heterogeneity of the tumor [65]. For this reason, we sometimes observe an upgrade of DCIS to IDC. AI could also have an important role in this kind of prediction. Qian et al. [73] used a DL algorithm to predict whether simple DCIS diagnosed with core needle biopsy would be upgraded to invasive cancer after surgical excision. The proposed model achieved good sensitivity, specificity, and accuracy (0.733, 0.750, and 0.742).

### 4.6. Predicting BC Prognosis

Classifications of molecular subtypes have a key role in the management strategy of BC. Several studies investigated the role of AI in BC prognosis, especially for TNBC. TNBC had a worse overall survival because of its higher nuclear grade, larger tumor size, and more aggressive proliferative index. Extracting 460 radiomic characteristics, Wang et al. [74] established an ML model for predicting disease-free survival in TNBC. In addition, Yu et al. [75], in a large multicenter study, assessed that radiomics is a promising biomarker for risk stratification for TNBC patients

## 5. Conclusions

US-based AI has great potential and research value in the field of breast cancer diagnosis, treatment, and prognosis. AI is considered a “hot topic” in radiology, and it has the potential to carry us into the era of personalized medicine. AI looks promising in every field of study evaluated in this review. More prospective and multicentric studies are needed to assess the potential impact of AI in breast ultrasound and to understand how to insert AI into the clinical workflow of radiologists.

## Figures and Tables

**Figure 1 diagnostics-13-00058-f001:**
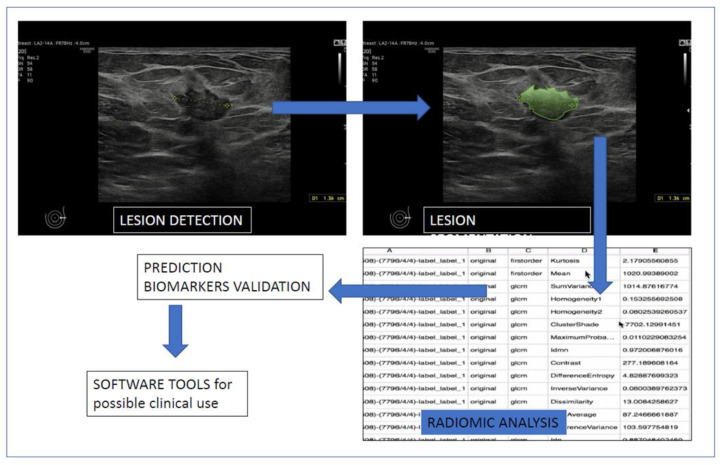
Flowchart: lesion detection, segmentation, validation, and test dataset.

**Figure 2 diagnostics-13-00058-f002:**
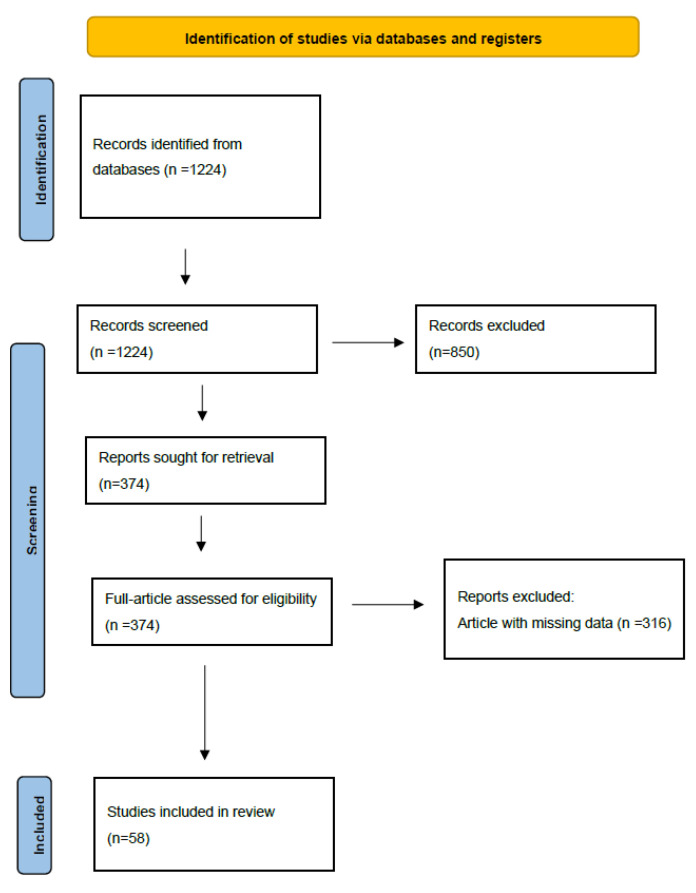
Flowchart of literature search.

**Table 1 diagnostics-13-00058-t001:** Specifications of studies on BC diagnosis included in this review.

Study	Aim	Modality	Design	Method	Test Dataset	Training and Validation Dataset	Performance
Niu et al. 2020 [5]	Diagnosis of BC among BI-RADS 4a lesions	US	Single-center retrospective	DL		206 images	AUC = 0.8746, ACC = 88.75%, SENS = 87.35%, SPEC = 90.32%
Fleury et al. 2019 [6]	Benign or malignant lesions	US	Single-center prospective	ML	206 lesions (144 benign, 62 malignant)		AUC = 0.840, SENS = 71.4%, SPEC = 76.9%
Romeo et al. 2021 [7]	Noncystic benign and malignant breast lesions	US	Multicenter retrospective	ML	174 patients, 201 lesions	135 patients	AUC = 0.82, ACC = 82%, SENS = 93%, SPEC = 57%, PPV = 82%, NPV = 80%
Lyu et al. 2022 [8]	Reduce the probability of BI-RADS 4 biopsy	US	Single-center prospective	DL	107 lesions		SENS = 93.48%, SPEC = 67.21%, ACC = 78.50%, VPP = 68.25%, VPN = 93.18%
Hayashida et al. 2019 [9]	Classify lesion in BI-RADS 3 or lower or BI-RADS 4a or higher	US,	Single-center retrospective	DL		3656 lesions	AUC = 0.95, SENS = 91.2%, SPEC = 90.7%
Fujioka et al. 2019 [10]	Distinguish between benign and malignant masses	US	Single-center retrospective	DL	48 benign masses, 72 malignant masses	480 images of 96 benign masses and 467 images of 144 malignant masses	SENS = 0.958, SPEC = 0.875, ACC = 0.925, AUC = 0.913
Wang et al. 2021 [11]	Reduce excessive lesion biopsy	US	Single-center prospective	DL	167 patients with 173 lesions		SENS = 94.9, SPEC = 69.4, AUC = 0.82, ACC = 80.9
Shen et al. 2021 [12]	Identify malignant lesions and reduce false positive examinations	US, CDFI	Multicenter retrospective	DL	5442907 images	4584271 images	AUC = 0.976 for internal test set, AUC = 0.927 for external test set
Huo et al. 2021 [13]	Distinguish between malignant and benign lesions	US	Multicenter retrospective	ML	1125 cases	1965 cases	AUC = 0.906, ACC = 0.772, precision = 0.528, recall = 0.936, F1-score = 0.675
Gao et al. 2021 [14]	Distinguish between malignant and benign lesions	US	Multicenter retrospective	DL	8966 nodules	5396 nodules	AUC = 0.934
Du et al 2022 [15]	Distinguish between malignant and benign lesions	US	Single-center retrospective	DL	1181 images	1031 images	ACC = 92.6%
Zhang et al. 2021 [16]	Establish an AI model	US	Single-center retrospective	DL	1311 lesions	1109 lesions	AUC = 0.846, ACC = 77.0%, SENS = 82.0%, SPEC = 71.7%, PPV = 79.3%, NPV = 75.1%
Wan et al. 2021 [17]	Distinguish between malignant and benign lesions	US	Retrospective	ML	895 images	800 images	AUC = 0.95, ACC = 0.86, SENS = 0.84, SPEC = 0.88
Shia et al. 2021 [18]	Distinguish and classify between malignant and benign lesions	US	Single-center retrospective	DL	677 images		AUC = 0.938, SENS = 94.34 %, SPEC = 93.22 %, PPV = 92.6%, NPV = 94.8%
Li et al. 2021 [19]	Radiologist performance with and without AI	US, SWE	Retrospective	DL	2967 images	2368 images	AUC = 0.892, ACC = 92.4%, SENS = 81.5%, SPEC = 96.9%
Zhang et al. 2020 [20]	Classify breast lesions	US, SWE	Retrospective	DL	291 masses	198 masses	US-AUC = 0.99, SWE-AUC = 0.99
Tanaka et al. 2019 [21]	Classify breast lesions	US	Multicenter retrospective	DL	1536 masses	1382 masses	AUC = 0.951, ACC = 89.0%, SENS = 90.9%, SPEC = 87.0%
Qi et al. 2019 [22]	Distinguish malignant tumors and solid nodules	US	Single-center retrospective	DL	8145 images	6786 images	AUC = 0.982, ACC = 94.48%, SENS = 95.65%, SPEC = 93.88%
Byra et al. 2019 [23]	Classify breast mass	US	Retrospective	DL	882 images	732 images	AUC = 0.936, ACC = 0.887, SENS = 0.848, SPEC = 0.897
Xiao et al. 2018 [24]	Differentiate benign and malignant tumors	US	Single-center retrospective	DL	2058 lesion		AUC = 0.93, ACC = 89.44%, SENS = 88.73%, SPEC = 89.91%
Sultan et al. 2018 [25]	Breast cancer diagnosis	US + CDFI	Single-center retrospective	ML	160 lesions		AUC = 0.89, SENS = 79%, SPEC = 89%
Becker et al. 2018 [26]	Classify breast cancer	US	Single-center retrospective	DL	637 lesions, 637 images	637 images	AUC = 0.84, SENS = 84.2%, SPEC = 80.3%, PPV = 32.0%, NPV = 97.9%
Gu et al. 2022 [27]	Differentiating benign from malignant breast lesions	US	Multicenter prospective	DL	14,043 US images	4149	AUC = 0.913, SENS = 88.84%, SPEC = 83.77%, ACC = 86.40%
Wei et al. 2022 [28]	Distinguishing benign and malignant breast masses	US	Multicenter prospective	ML	901 breast masses		ACC = 89.0%, SENS = 91.4%, SPEC = 87.7%, PVV = 80.8%, NPV = 94.7%
Wilding et al. 2022 [29]	Benign and malignant classes	US	Single-center retrospective	ML	953 masses		ACC = 96%
O’Connell et al. 2022 [30]	Assist radiologists in the diagnosis of breast cancer	US	Multicenter retrospective	DL	299 lesions		SENS, SPEC and ACC greater than 0.8
Ma et al. 2021 [31]	Assist radiologists in the diagnosis of benign and malignant tumors	US	Single-center retrospective	DL	952 lesions	1052 images	ACC = 92%, SENS = 95.65%, SPEC = 88.89%, AUC = 0.97
Chowdhury et al. 2022 [32]	Assist in the diagnosis by helping to reduce false positive diagnoses	US	Retrospective	ML			ACC = 93.08%, AUC = 0.9712, FNR = 0%, FPR = 8.65%
Li et al. 2021 [33]	Distinguish benign from malignant lesions	US	Multicenter retrospective	DL	692 images	271 malignant nodules, 1053 benign nodules and 2144 images of the contralateral normal breast	AUC = 0.943, SENS = 73.3%, SPEC = 94.9%
Lai et al. 2022 [34]	Compare the diagnostic performance between reading without AI	US	Single-center retrospective	DL	172 lesions		Increase in the average AUC ROC from 0.7582 to 0.8294
Li et al. 2022 [35]	Enhance the detection accuracy with AI system	US	Single-center retrospective	DL	156 images	624 images	Accuracy benign = 0.651 Accuracy malignant = 0.579
Zhang et al. 2020 [36]	Prediction of breast cancer	US	Single-center retrospective	DL	1007 images	5000 images	AUC = 0.913

**Table 2 diagnostics-13-00058-t002:** Specifications of studies on prediction of molecular subtypes of BC included in this review.

Study	Aim	Modality	DESIGN	Method	Test Dataset	Training and Validation Dataset	Performance
Ye et al. 2021 [37]	Identify TNBC from non-TNBC	US	Single-center retrospective	DL		204 images (102 NTN and 102 TN)	AUC = 0.8746, ACC = 88.75%, SENS = 87.35%, SPEC = 90.32%
Zhang et al. 2021 [38]	Predict molecular subtypes	US	Multicenter retrospective	DL	1235 images, 790 patients	Patients/images: 684/986 HR+ = 417/588, HER2+ = 149/212, TNBC = 118/186	AUC 0.864, 0.811 and 0.837 for TN, HER2 (+) and HR (+) subtype
Xu et al. 2022 [39]	Predict the expression of HER2	US	Single-center retrospective	DL	144 patients	144 patients	AUC = 0.84, ACC = 80.56%, SENS = 72.73%, SPEC = 84.00%, PPV = 66.67%, NPV = 87.5%
Wu et al. 2019 [40]	Identify TNBC from non-TNBC	US	Single-center retrospective	ML	140 images	131 patients	AUC = 0.88, SENS = 86.96%, SPEC = 82.91%
Zhou et al. 2021 [41]	Predict four- and five-classification molecular subtypes	US, SWE, CDFI	Multicenter prospective	DL	807 patients, 818 images	534 patients, 545 breast cancers	Test A: AUC = 0.92, SENS = 91.67%, SPEC = 78.57%, ACC = 81.72% Test B: AUC = 0.96, SENS = 87.50%, SPEC = 87.50%, ACC = 82.11%
Ma et al. 2022 [42]	Predict molecular subtypes	Mammography, US	Single-center retrospective	ML	600 patients	450 patients	TNBC vs. other: AUC = 0.971, ACC = 0.947, SENS = 0.905, SPEC = 0.941 Luminal vs. other: AUC = 0.900, ACC = 0.860, SENS = 0.871, SPEC = 0.886 HER2 vs. other: AUC = 0.855, ACC = 0.893, SENS = 0.900, SPEC = 0.724 ER-positive vs. ER-negative: AUC = 0.878, ACC = 0.867, SENS = 0.940, SPEC = 0.788
Guo et al. 2018 [43]	Assess biological characteristics	US	Single-center retrospective	ML	215 patients		SENS = 0.979, SPEC = 0.601, AUC = 0.760
Jiang et al. 2021 [44]	Luminal disease from non luminal disease	US	Multicenter retrospective	DL	1132 images, 845 patients	4828 US images from 1275 patients	Test A: Luminal A: AUC = 0.89, ACC = 86.96% Luminal Test B: AUC = 0.86, ACC = 80.07% HER2+: AUC = 0.80, ACC = 90.88% TNBC: AUC = 0.82, ACC = 97.02%

**Table 3 diagnostics-13-00058-t003:** Specifications of studies on prediction of axillary lymph node metastases in BC included in this review.

Study	Aim	Modality	Design	Method	Test Dataset	Training and Validation Dataset	Performance
Zheng et al. 2020 [45]	Predict ALN status (N0-N1)	US	Prospective	DL	584 lesions	466 lesions	AUC = 0.902, ACC = 81.0%, SENS = 81.6%, SPEC = 83.6%, PPV = 78.4%, NPV = 86.2%
Jiang et al. 2021 [46]	ALN burden in early-stage breast cancer	US, SWE	Multicenter retrospective	DL		303 + 130 patients	C-index of 0.845 for the training cohort and 0.817 for the validation cohort
Zhou et al. 2020 [47]	Clinically negative axillary lymph node metastasis	US	Multicenter retrospective	DL	834 patients, 1055 images	680 patients, 877 images	AUC = 0.89, ACC = 79%, SENS = 85%, SPEC = 73%, PPV = 76%, NPV = 81%
Tahmasebi et al. 2021 [48]	Benign and malignant ALN	US	Single-center retrospective	ML	317 images	64 images	AUC = 0.78, ACC = 69.5%, SENS = 74.0%, SPEC = 64.4%, PPV = 68.3%, NPV = 72.6%
Ozaki et al. 2022 [49]	Normal and metastatic axillary lymph node	US	Single-center retrospective	DL	300 images normal and 328 images metastatic	100 images (50 normal, 50 metastases)	SENS = 94%, SPEC = 88%, AUC = 0.966
Guo et al. 2020 [50]	Predict the risk of sentinel LN and nonsentinel LN metastasis	US	Multicenter retrospective	DL	3049 images	542 patients (SLN), 180 patients (NSLN)	SLN: AUC = 0.848, VPP = 61.5%, VPN = 97% NSLN: AUC = 0.81, VPP = 78.55%, VPN = 91.7%
Lee at al. 2021 [51]	Predict the axillary lymph node (ALN) metastatic status in patients with early-stage breast cancer	US	Single-center retrospective	DL	153 patients	153 patients	AUC = 0.805, ACC = 81.05%, SENS = 81.36%, SPEC = 80.85%, PPV = 72.73%, NPV = 87.36%
Sun et al. 2022 [52]	Predict ALN metastasis	US	Single-center retrospective	DL	338 images	248 images	AUC = 0.72, ACC = 72.6%, SENS = 65.5%, SPEC = 78.9%

**Table 4 diagnostics-13-00058-t004:** Specifications of studies on prediction of response to NAC included in this review.

Study	Aim	Modality	Design	Method	Test Dataset	Training and Validation Dataset	Performance
DiCenzo et al. 2020 [53]	Assess pCR to NAC	US, QUS	Multicenter prospective	ML	82 patients		SENS = 91.2, SPEC = 83.3, AUC = 0.726
Jiang et al. 2021 [54]	Assess pCR to NAC	US	Multicenter retrospective	DL	592 patients	356 patients	SENS = 89.3%, SPEC = 81.3%, AUC = 0.94
Gu et al. 2021 [55]	Response to NAC after the second course and fourth course of therapy	US	Single-center prospective	DL (two different models)	592 patients	356 patients	DLR-2: AUC = 0.812, ACC = 69.1%, SENS = 90.5%, SPEC = 47.6%, NPV = 83.3% DLR-4: AUC = 0.937, ACC = 85.7%, SENS = 81.0%, SPEC = 90.5%, NPV = 82.6%
Taleghamar et al. 2022 [56]	Responders and nonresponders before starting therapy	QUS	Single-center retrospective	DL	181 patients	131 patients	AUC = 0.86, ACC = 88%, SENS = 70%, SPEC = 92.5%
Xie et al. 2022 [57]	Predict pCR before and after the first stage of NAC	US	Single-center retrospective	DL	114 patients, 968 images	776 images	AUC = 0.939, ACC = 87.50%, SENS = 90.67%, SPEC = 85.67%, PPV = 80.00%, NPV = 93.46%
Byra et al. 2021 [58]	Responders and nonresponders	US	Single-center retrospective	DL	39 tumors		After the second NAC: AUC = 0.847, ACC = 76.9%, SENS = 78.9%, SPEC = 75.2%

## Data Availability

The datasets generated during and/or analyzed during the current study are available from the corresponding author on reasonable request.

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
