# Peer review of "Artificial Intelligence in Breast Ultrasound: From Diagnosis to Prognosis—A Rapid Review"

_diagnostics, 2022, doi:10.3390/diagnostics13010058_

Round 1

Reviewer 1 Report

The authors made a great effort to conduct this research work. I have the following comments.

1.   Line 66, 67: ‘ultrasound s’, ‘breast s’

2.   Check brackets carefully in search strategy.

3.   Line 71, 72: whether the conference papers are excluded?

4.   Line 78: tell ‘possible clinical use’ into more detail.

5.   Lack of PRISMA flowchart of systematic search and selection process.

6.   Need more details of deep learning and machine learning techniques. For example, the models or how is ‘image processing, segmentation and feature extraction’ (Line 40) implemented?

7.   Discuss the comparison of ML and DL, such as, capabilities, limitation and applicable scenarios, etc.

8.   The discussion section contains too much unnecessary repeat of the information from tables.

9.   Line 152: discuss the meaning of the average values of sensitivity, specificity, AUC.

10. Lack of summary of ‘the current development and research status of US-based AI in the field of breast cancer’ as mentioned in the aim.

11. Whole field of ultrasound elastography is missing. Some recent articles should be included for reference:

Mao, Y.-J.; Lim, H.-J.; Ni, M.; Yan, W.-H.; Wong, D.W.-C.; Cheung, J.C.-W. Breast Tumour Classification Using Ultrasound Elastography with Machine Learning: A Systematic Scoping Review. Cancers 2022, 14, 367. https://doi.org/10.3390/cancers14020367

Zheng X, Yao Z, Huang Y, et al. Deep learning radiomics can predict axillary lymph node status in early-stage breast cancer [published correction appears in Nat Commun. 2021 Jul 12;12(1):4370]. Nat Commun. 2020;11(1):1236. Published 2020 Mar 6. doi:10.1038/s41467-020-15027-z

Kim MY, Kim SY, Kim YS, Kim ES, Chang JM. Added value of deep learning-based computer-aided diagnosis and shear wave elastography to b-mode ultrasound for evaluation of breast masses detected by screening ultrasound. Medicine (Baltimore). 2021;100(31):e26823. doi:10.1097/MD.0000000000026823

12.  There are no machine learning or deep learning or learning keywords involved in the search key

13.  Search should perform in more than one database such as Web of Science (topic field, articles, English), CINAHL via EBSCOhost (default field) and EMBASE via OVID (topic field, English).

14.  Algorithm exploited will cause significant impact of performance. Algorithms used in the studies should be listed.

Author Response

Reviewer #1:

Comment 1: Line 66, 67: ‘ultrasound s’, ‘breast s’

R: thank you for the suggestion. We correct the typing errors.

Comment 2: Check brackets carefully in search strategy.

R: thank you for the suggestion. We checked carefully brackets in the search strategy.

Comment 3 Line 71, 72: whether the conference papers are excluded?

R: yes, we only considered original articles.

Comment 4 Line 78: tell ‘possible clinical use’ into more detail.

R: thank you for the suggestion. We specified the possible clinical uses, such as diagnosis, reduction of biopsies number or as prognostic tools.

Comment 5 Lack of PRISMA flowchart of systematic search and selection process.

R: thank you, we added PRISMA flow chart as suggested.

Comment 6  Need more details of deep learning and machine learning techniques. For example, the models or how is ‘image processing, segmentation and feature extraction’ (Line 40) implemented?

R: thank you. We implemented this topic in manuscript.

Comment 7 Discuss the comparison of ML and DL, such as, capabilities, limitation and applicable scenarios, etc.

R: thank you. As requested we added to the manuscript some information regarding ML and DL.

Comment 8 The discussion section contains too much unnecessary repeat of the information from tables.

R: thank you for the suggestion. We removed some information from the discussion.

Comment 9  Line 152: discuss the meaning of the average values of sensitivity, specificity, AUC.

R: thank you. We added some information to the paper.

Comment 10 Lack of summary of ‘the current development and research status of US-based AI in the field of breast cancer’ as mentioned in the aim.

R: thank you for the clarification. We added sentences in the conclusion paragraph explain that AI present excellent performances in every fields analyzed in this review.

Comment 11 Whole field of ultrasound elastography is missing. Some recent articles should be included for reference:

R: thank you for the suggestion. We added this article to references as suggested.

Comment 12: There are no machine learning or deep learning or learning keywords involved in the search key

R: thank you for the suggestion. We used the term “artificial intelligence” in the search key, including both machine learning and deep learning.

Comment 13: Search should perform in more than one database such as Web of Science (topic field, articles, English), CINAHL via EBSCOhost (default field) and EMBASE via OVID (topic field, English)

 R: thank you for the suggestion. We also performed the research with EMBASE and Web of Science. We obtain the same results without discrepancies.

Comment 14: Algorithm exploited will cause significant impact of performance. Algorithms used in the studies should be listed.

R: we are grateful for the comment. We didn’t report the used algorithms because they weren't reported in all studies.

Reviewer 2 Report

Thanks for your manuscript. Here are some points about it: 

1. Between lines 58 and 61, you mention that your "search" was done by two experienced people in the field of your research." Please, could you validate your "search" in your work with other tools like, e.g., Prisma

2. In table 1 (page 5), the last row has this number ",2058 lesion". Please, verify it if this value is correct. 

Many thanks for considering my request.

Author Response

Dear Editor and Reviewers,

we would like to thank the Editor and the Reviewers for the thorough revision and help in improving the present manuscript. Please find below the changes that we made to improve the overall quality of the manuscript. We hope to have further improved both the readability of the manuscript and the scientific quality. We appreciated the constructive comments and we revised the manuscript according to the suggestions.

Reviewer #2:

Comment 1: Between lines 58 and 61, you mention that your "search" was done by two experienced people in the field of your research." Please, could you validate your "search" in your work with other tools like, e.g., Prisma

R: thank you, we added PRISMA flow chart as suggested.

Comment 2: In table 1 (page 5), the last row has this number ",2058 lesion". Please, verify it if this value is correct.

R: thank you for the suggestion. We checked and the value is correct.

Round 2

Reviewer 1 Report

Thanks for the update.